# Cutaneous Findings in Neurofibromatosis Type 1

**DOI:** 10.3390/cancers13030463

**Published:** 2021-01-26

**Authors:** Bengisu Ozarslan, Teresa Russo, Giuseppe Argenziano, Claudia Santoro, Vincenzo Piccolo

**Affiliations:** 1Dermatology Unit, Doku Medical Center, 34381 Istanbul, Turkey; bengisuozarslan@gmail.com; 2Dermatology Unit, University of Campania Luigi Vanvitelli, 80100 Naples, Italy; russo.teresa87@gmail.com (T.R.); giuseppe.argenziano@unicampania.it (G.A.); 3Department of Woman, Neurofibromatosis Referral Centre, Child and of General and Specialised Surgery, University of Campania Luigi Vanvitelli, 80100 Naples, Italy; claudia.santoro@unicampania.it

**Keywords:** neurofibromatosis type 1, skin, cutaneous, café-au-lait macules, neurofibroma, freckling, nevus anemicus, juvenile xanthogranuloma

## Abstract

**Simple Summary:**

Neurofibromatosis type 1 (NF1) is characterized by major and minor cutaneous findings, whose recognition plays a key role in the early diagnosis of the disease. The disease affects multiple systems and clinical manifestation has a wide range of variability. Symptoms and clinical signs may occur over the lifetime, and the complications are very diverse. Although significant progress has been made in understanding the pathophysiology of the disease, no specific treatment has been defined. Multidisciplinary approach is required to provide optimum care for the patients. The aim of this paper is to provide the clinician with a complete guide of skin findings of NF1.

**Abstract:**

Neurofibromatosis type 1 (NF1) is a complex autosomal dominant disorder associated with germline mutations in the NF1 tumor suppressor gene. NF1 belongs to a class of congenital anomaly syndromes called RASopathies, a group of rare genetic conditions caused by mutations in the Ras/mitogen-activated protein kinase pathway. Generally, NF1 patients present with dermatologic manifestations. In this review the main features of café-au-lait macules, freckling, neurofibromas, juvenile xanthogranuloma, nevus anemicus and other cutaneous findings will be discussed.

## 1. Background

Neurofibromatosis type 1 (NF1) is a complex autosomal dominant disorder associated with germline mutations in the NF1 tumor suppressor gene [1]. NF1 belongs to a class of congenital anomaly syndromes called RASopathies, a group of rare genetic conditions caused by mutations in the Ras/mitogen-activated protein kinase (RAS/MAPK) pathway [2]. Generally, NF1 patients present with dermatologic manifestations (Table 1). Other abnormalities in the nervous system and skeletal system may also develop [3].

## 2. Genetic

NF1 is inherited in an autosomal dominant pattern, and a single copy of a mutated or deleted *NF1 gene* is needed in order to be effective. Half of the cases are familial, but the remaining half arise from de novo NF1 mutations. Descendants of an NF1 patient are at a 50% risk of inheriting the altered *NF1 gene*; however, the disease manifestations vary extremely even within the same family members. In addition, somatic mutations in NF1 during fetal development also cause phenotype diversity, known as mosaic NF1 [4].

*NF1 gene* encodes neurofibromin protein which interacts with the Ras-specific guanosine triphosphatase (GTPase), a family of proteins involved in cellular signal transduction, controlling cell proliferation. Overactive Ras-GTPase signals lead to tumor growth. Neurofibromin has the role of downregulating the RAS/MAPK signaling pathway and suppressing cell overgrowth. Therefore, the *NF1 gene* is considered to be a tumor suppressor gene and has been associated with benign and malignant tumors in many different areas from nerve cells to myeloid cells [5].

A diagnosis of NF1 is clinically established by the presence of two or more of the following diagnostic criteria as formulated by the National Institute of Health (NIH) in 1988: six or more cafe-au-lait macules (CALMs) (>5 mm in children or >15 mm adults), two or more cutaneous neurofibromas (CNFs) or one plexiform neurofibroma (PNF), axillary or inguinal freckling, optic nerve glioma (OPG), two or more Lisch nodules, distinctive bony dysplasia (sphenoid wing dysplasia and typical bowing of long-bone with or without pseudo-arthrosis), and first degree relative with NF1 [6]. Although it is rarely needed after early childhood, genetic testing may be required primarily in patients who do not meet the diagnostic criteria. It is also important in situations where the rapid diagnosis may affect the course of the disease and the treatment (children with a serious tumor). Additionally, a severe NF1 variant with spinal cord neurofibromas has been identified in several studies. In the lack of other diagnostic features, genetic testing is indicated to diagnose these patients and at-risk family members. Finally, prenatal or preimplantation genetic testing may be necessary for the offspring of NF1 parents [4]. However, performing genetic analyses and genotypic–phenotypic correlations are not easy for several different reasons. The *NF1 gene,* which has been identified so far as having more than 3000 pathological allele variants, is a large gene covering 275 kb of genomic DNA and containing almost 60 constitutive exons located at 17q11.2 [7]. There may be missense mutations/in-frame deletions, truncating/splicing mutations and large deletions in the gene. The absence of defined mutation hot spots makes it necessary to analyze every coding exon. Numerous NF1 pseudogenes with high sequence homologies and deep intronic splicing mutations further complicate molecular analyses of the *NF1 gene* [8,9]. There are different methods used for genetic analysis, of which the most accurate is the multistep pathogenic variant detection protocol based on cDNA (mRNA) and genomic DNA sequence analysis. In this method, diagnostic sensitivity has been reported to be more than 95% [10].

The clinical presentation of NF1 may be multifactorial. While biallelic inactivation is necessary for the development of some clinical findings such as CALMs and neurofibromas, some manifest in haploinsufficiency of NF1. Additional genetic alterations also affect the disease course as in the malignant transformation of peripheral nerve sheath. Even though clear genotype–phenotype correlations, which are hard to establish in NF1, there are some important relations identified. One study investigating mutation and phenotype trends found that NF1 was more prevalent in patients with truncating/splicing mutations and large deletions than in those with missense mutations. They also reported more items of NF1 in patients with truncating/splicing mutations [9]. Another study showed an association between 1.4 Mb deletions that encompass the entire *NF1 gene* with nonfamilial facial dysmorphism, a lower IQ and possibly increased incidence of malignant peripheral nerve sheath tumors (MPNSTs). Codeletion of *SUZ12,* especially, within the common NF1-microdeletion has been identified to be a significant alteration for the malignant transformation [11]. Noonan-like features (pulmonic stenosis and short stature) and multiple CALMs, but no external PNF or symptomatic OPG have been linked to NF1 missense variations affecting *p*.Arg1809 [12]. A small mutation in exon 17 of the *NF1 gene* identified as a single amino acid NF1 deletion (c.2970–2972 delAAT) has been related to mild form of NF1 disease without visible CNFs or PNFs [13]. Koczkowska et al. reported a severe form of NF1 presenting as major PNF and symptomatic spinal neurofibromas, OPGs, other malignant transformations and skeletal abnormalities due to missense mutations affecting NF1 codons 844–848 [14]. Further investigations are needed to clarify the genetic background of NF1.

## 3. Mosaicism

Mosaic NF1 is caused by postzygotic somatic mutation in the *NF1 gene*. It occurs usually with no antecedent family history. NF1 in mosaic forms can be present as generalized, segmental, or gonadal [15]. Early somatic mutations that occur before tissue differentiation manifest as a generalized phenotype. The generalized mosaic pattern appears clinically indistinguishable from classical nonmosaic NF1 but the symptoms tend to be milder. Blood leucocyte screening usually does not reveal the specific NF1 mutation. Late somatic mutations cause a limited distribution, classified as segmental NF1. Segmental NF1 has been divided into four groups as follows: pigmentary alterations (CALMs and freckling) alone, CNFs alone, pigmentary alterations plus neurofibromas and isolated PNFs. Skin manifestation is limited to the affected area, which may cover either a narrow strip or one quadrant sometimes covers half of the body. Classically, it is unilateral, but it can be bilateral as well. CALMs may be distributed in a block-like or blaschkoid pattern whereas neurofibromas usually follow the dermatomes. A darker background may cover the affected region [16]. Maertens et al. showed biallelic inactivation of the *NF1 gene* in either the Schwann cells of neurofibromas or in the melanocytes of CALMs in individuals with different types of segmental disease [17]. While it is known that the gonadal type can affect the offspring, segmental NF1 cases may be transmitted to future generations with some unknown mechanisms [15] (Figure 1).

## 4. Cafe-Au-Lait Macules

A CALM is a pigmented, flat skin patch which is the most common initial manifestation of NF-1. CALM is seen in 99% of NF1 patients. The classical CALM lesion in NF1 has a round or oval shape with a uniform color of light to dark brown and regular well-demarcated margins (“coast of California”) [18]. CALMs may be observed at birth, become almost always obvious by the first year of life, and may increase in diameter and number. NF1 patients have higher numbers of CALMs widely distributed on any part of the body except for palms and soles. CALMs rarely occur on the face. Although one to three macules may be present in up to 15% of the normal population, six or more of these lesions greater than 0.5 cm in prepubertal children or greater than 1.5 cm after puberty are diagnostics for NF1 [6]. However, it is also recommended to evaluate and follow the patients with more than three CALMs, as it is less frequent to see more than three lesions in the normal population [19]. There is no relation between NF1 severity and number of lesions [20]. Macules may get tanned under UV exposure and Wood′s lamp may be helpful to detect CALMs invisible to the naked eye [21]. Underlying PNF may accompany CALMs, especially those with hypertrichosis. Therefore, palpation is recommended [22] (Figure 2, Figure 3, Figure 4 and Figure 5).

Another type of CALM which has irregular jagged margins (“coast of Maine”) is mostly solitary and larger; arrangement in a blaschkoid pattern is suggestive of sporadic cases or other genetic disorders. Additionally, it is extraordinary for new sporadic CALMs to occur after the age of 6 [18].

Histological findings are basal hyperpigmentation and sometimes giant melanosomes (macromelanasomes), although diagnosis is clinical and pathology rarely necessary. Recently, mutation of both alleles of the NF1 gene has been detected in the melanocytes of NF1-CALM while normal skin has a germline mutation [23]. CALMs in NF1 tend to show greater melanocyte density and greater stem-cell factor secretion (KIT ligand) of fibroblasts while non-NF1 CALM lesions show the same melanocyte level compared to the normal skin. A melanocyte activation response to hepatocyte growth factor and KIT ligand has been hypothesized as the mechanism of melanocyte growth and mast cell infiltration [24,25].

## 5. Freckling

Freckling in skinfold is one of the seven cardinal diagnostic features of NF1. They are described as small (1–4 mm), clustered pigmented macules located mostly in axillary and inguinal regions (Crowe sign) [26]. It is considered pathognomonic and found in more than 90% of NF1 patients by the age of 7. The development of freckles may not be seen as early CALMs. They often occur by the age of three to five following CALMs and precede the appearance of neurofibromas [27]. Typically, these lesions resembling sun-induced freckles appear first in the inguinal region [28]. The presence of only one or two small freckles—or freckling of other intertriginous areas such as the neckline, upper eyelid, around the lips, under breasts in women, even a more diffuse pattern over the body may arise suspicion—are not sufficient for diagnosis [15] (Figure 6 and Figure 7).

## 6. Lisch Nodules

Lisch nodules are benign iris hamartomas which are seen as densely pigmented flat or slightly elevated lesions during the slit-lamp examination. These lesions are the most common NF1 feature in adults. They appear earlier than neurofibromas and multiple Lisch nodules are specific to NF1. They do not have any ophthalmologic complication, no association has been reported with clinical severity, and therefore no specific treatment is needed. However, periodical examination to detect these benign hamartomas may be useful to confirm the diagnosis of young NF1 patients [29].

## 7. Neurofibromas

Neurofibromatosis is associated with many kinds of tumors but the neurofibromas are the most prevalent benign tumors found in the affected individual, hence the appellation neurofibromatosis. They also have a negative impact on the life of NF1 patients. Different variants have been described related to NF1; CNFs or dermal neurofibromas, subcutaneous neurofibromas and PNFs [21,30]. While loss of the neurofibromin allele in Schwann cells is known to be the initial trigger, their stem cell factor secretion drive NF1-gene-deficient mast cells into the neurofibroma lesion. Migration of mast cells leads to tumor formation as they express transforming growth factor-β (TGFβ). TGFβ boosts Nf1-deficient Schwann cell growth and collagen production. Generally, proliferation of spindle cells, Schwann cells, mast cells and vascular components leads to manifestation of these tumors [3,31,32].

Thousands of discrete CNFs, fleshy, pink to brown pedunculated or sessile papulonodules, depict the most common dermatological finding in NF1. They may be soft or firm in texture ranging from few millimeters to few centimeters located anywhere along the course of peripheral nerves. “Buttonhole sign” describes the pathogonomic invagination of the lesion with pressure. Subcutaneous neurofibromas arise as nodules in deep dermis or the subcutaneous layer, usually presenting a less obvious appearance than CNFs. Both cutaneous and subcutaneous neurofibromas may initially appear as slightly raised discolored skin patches and most typically they become visible in adolescence. They may grow and increase in number with age; acceleration may occur during pregnancy [15] (Figure 8, Figure 9, Figure 10 and Figure 11).

Blue red macules and pseudoatrophic macules are bizarre variants of cutaneous neurofibroma. If there is the predominance of wide vessels in neurofibromatous tissue located in papillary dermis, the lesion may appear as blue-red macules. Pseudoatrophic macules occur as a result of neurofibromatous tissue proliferation damaging collagen in reticular dermis. These two presentations are less frequent types of neurofibromas and they fulfill the diagnostic criterion of NF1 [33].

Physical removal appears to be the most effective and practical treatment for CNFs as there is no gold standard approach. Surgical removal, laser ablation, modified biopsy removal, photocoagulation, electrodessication and radiofrequency ablation are the modalities that have been developed. Camouflage and moisturizers may also provide some relief for patients [34]. In a previous study, topical imiquimod has shown some volume reduction. Since vascular endothelial growth factor (VEGF) is known to be highly expressed in neurofibromas, Ranizumab, a VEGF antibody, was injected into a tumor and a decrease in volume and pressure was noticed [35].

PNFs are specific for NF1 and they affect between 30% and 50% of patients [36,37]. PNFs arise from the peripheral nerve sheath. They are often located along the nerve and may be superficial or may develop into a complex deep mass with the network of multiple nerves. PNFs may appear as thickened, slightly raised skin areas or firm masses or nodules in the subcutaneous tissue. Lesions may also be present with a particular wavy “bag of worms” texture [38]. Deep lesions may infiltrate structures such as fascia, muscle or more internal components of the body causing disfigurement and dysfunction [36]. Unlike CNFs, PNFs are congenital and superficial lesions and may be clinically visible at birth, resembling CALMs. Those with hyperpigmentation and hypertrichosis should be distinguished from congenital melanocytic nevus and smooth muscle hamartoma Becker nevus [22]. Magnetic resonance imaging (MRI) may be needed to detect deep, hidden lesions. They may expand during childhood and adolescence but after they grow at a fairly low rate [39].

PNFs are associated with increased morbidity and mortality in NF1. Distortion of limbs or head, soft tissue and bone deformities, degeneration of airway or spinal cord and neurological deficits may occur due to invasion [40]. Spinal neurofibromas, particularly C2 nerve root neurofibromas, tend to occur bilaterally, with high incidence of intradural invasion and myelopathy in NF1 patients. The higher frequency of C2-located neurofibromas in patients with NF1 was attributed to the risk of trauma. There are hypotheses regarding increased post-traumatic fibroblast activity or “second hit” mutations in the *NF1 gene* [41]. As we mentioned before, there is also a spinal NF1 subtype which is associated with large deletions in the *NF1 gene*. Spinal NF1 is characterized by multiple bilaterally located spinal and paraspinal neurofibromas with very few dermal manifestations of classical NF1. This form should be differentiated due to life-threatening massive neoplastic burden [42]. Although the treatment of large tumors causing progressive neurological deficit includes surgery, the nonsurgical management remains a clinical challenge [43]. The RAF/MEK/ERK pathway is an important regulator of Schwann cell biology and several biologically targeted therapies (such as mTOR inhibitors, imatinib and selective MEK inhibitors) have been evaluated in clinical trials. Selumetinib, an oral selective inhibitor, partially contributed to reduce PNF tumor size [44].

## 8. Malignant Peripheral Nerve Sheath Tumors

MPNSTs are aggressive sarcomas which may develop in 3–15% of patients with NF1 [45]. They mostly transform from plexiform or nodular neurofibromas predominantly after puberty. Especially in NF1 affected young adults, the rapid expansion of the tumor with pain or the presence of new neurological symptoms should alert the clinician for MPNST [46]. Coffien et al. reported NF1 patients with MPNST occurring as a second malignant neoplasm especially after alkylating agent chemotherapy and irradiation [47]. Positron emission tomography imaging is shown to be a more useful diagnostic tool to reveal these malignant degenerations [40].

Although the inactivation of the loss of biallelicity of the *NF1 gene* in Schwann cells seems to be an important initiating factor in the development of MPNST, genomic studies show that the malignant degeneration arises as a result of some additional alterations. Somatic loss-of-function mutations or deletions of the Polycomb repressive complex 2 (PRC2) its core subunits (EED or SUZ12) have been described in NF-1-associated MPNSTs. PRC2 plays a role in chromatin compaction, cell proliferation, differentiation and stem-cell plasticity. A cooperative role of TP53, epidermal growth factor receptor (EGFR) inactivation has also been proposed to contribute MNPST transformation. CDKN2A loss has also been linked to early steps of MPNST [48].

While high-grade MPNST often metastasizes to bone and lung, low-grade tumors have a better prognosis. The main therapy remains a complete surgical excision with negative margins [49]. Discussions on the benefits of adjuvant/neoadjuvant chemotherapies continue. Epirubicin-based adjuvant/neoadjuvant chemotherapy appears to show some survival benefit [50]. Although the development of radiotherapy-induced neoplasms is known to be increased in NF1 patients, radiotherapy may be useful for downsizing the tumors. On the other hand, research shows that radiotherapy benefits local control of the tumor without contributing to survival [51].

Metastatic disease remains challenging due to a lack of promising therapies. Chemotherapy is known to be the standard approach. Biological therapies combining MEK/mTOR inhibition are still under investigation in preclinical and clinical trials. Pexidartinib (tyrosine kinase inhibitor) and sirolimus (mTOR inhibitor) combination contributed to stabilize the disease in a phase I study [52]. Additionally, immunotherapy has also opened up a method for the treatment of MPNST [53].

In terms of the future direction of the treatments, gene-based therapies targeting PRC2, the Janus Kinase (JAK) pathway and HIPPO pathway are presented as a new hope. EZH2 inhibitors—such as tazemetostat for the PRC2 complex, Tyrosine Kinase 2 (TYK2) inhibitors targeting the JAK/STAT pathway and verteporfin inhibiting the HIPPO pathway—could be considered in the combination treatments [54].

## 9. Optic Pathway Glioma

NF1 is one of the most common heritable brain tumor predisposition disorders, in which affected individuals develop low-grade gliomas classified as WHO grade I gliomas (pilocytic astrocytomas). They are generally indolent neoplasms usually occurring before the age of 8 located within the optic pathway [55]. OPGs occur in 15–20% of NF1 patients [56]. They are often asymptomatic until the age of 7. The tumor may arise from anywhere along the optic pathway such as nerves, chiasm, postchiasmatic tracts, and radiations. Tumor location strongly affects the symptom spectrum. While 30–50% of OPGs become symptomatic, patients may present with unilateral proptosis, visual acuity loss, visual field defect, strabismus, relative afferent pupillary defect, and optic papilloedema or optic nerve atrophy. Chiasmal optic pathway gliomas can present with precocious puberty without any visual symptoms. Rarely, headache and vomiting may be signs of tumors involving the hypothalamus. Children usually do not complain of visual defects and NF1-OPGs can be unpredictable—as a result, a routine eye examination is required [57].

Although prognostic features are not clear, clinical experiences show that girls experience visual impairment more than boys. Additionally, postchiasmatic optic pathway-located OPGs and tumors that occur before the age of 2 appear to be more aggressive. After diagnosis, performing an eye examination and visual assessment with MRI is requested every 3 months in the first year for follow up.

Chemotherapy remains a mainstay treatment. Surgery seems to be problematic due to the location of these tumors and radiation is known to be a trigger for secondary malignancies along with vasculopathy. mTOR and MEK inhibitors for the treatment of optic pathway gliomas are still in progress [58].

## 10. Nevus Anemicus

Nevus anemicus (NA) is a congenital, cutaneous anomaly characterized by hypopigmented, well-defined patches with limited vascular supply. While rubbing or heat causes redness of normal skin, the NA lesion remains pale. The vascular structure of the lesion is found to be normal but the excessive response to catecholamines causes vasoconstriction. NA is also seen in the normal population, but prospective studies suggest that prevalence of NA in NF1 patients is much higher—around 50%, affecting both sexes equally [59,60,61]. The lesions are usually multiple and mostly located on the trunk in varying sizes. The age of onset is unclear as it may be difficult to detect the lesion clinically. The link between NA and NF is not well-established. NF is associated with vascular complications and it has been postulated that there might be a somatic or second-hit mutation in NA lesions [62] (Figure 12).

## 11. Juvenile Xanthogranuloma

Juvenile xanthogranuloma (JXG) is the most common form of non-Langerhans-cell histiocytosis characterized by yellowish-pink dome-shaped granulomatous tumors often located on the upper side of the body. It may be solitary or multiple and may start as a pink macule. The estimated prevalence is around 15–35% in NF1 children. JXG may be present as early as CALM, therefore it was suggested as a potential sign for the diagnosis of NF1. The lesion may disappear spontaneously [61,63]. Juvenile myelomonocytic leukemia (JMML) is a rare, fatal disorder of childhood [64]. It has been suggested that NF1 patients with JXG have a considerably higher risk of developing JMML than those with only NF1 [65]. There is no clear consensus on the increased risk of JMML in the presence of JXG in NF1 patients since other retrospective case series did not reveal such association [66,67]. Although routine screening for JMML in NF1 patients is not obligatory, the physicians should be careful about the other signs of JMML (Figure 13).

## 12. Glomus Tumor

Glomus tumor is a benign neoplasm that arises from the neuromyoarterial glomus body or the Suquet–Hoyer canal. Lately, the discovery of biallelic inactivation of the *NF1 gene* or a “second hit” that occurs specifically within the α-smooth muscle actin-positive cells of the glomus body has been added to an NF1-related tumor spectrum [68]. It is often located at the subungual area, clinically presented as a small reddish-blue papulonodule that is particularly painful after pressure or temperature change. Regarding the location, it can arise from almost everywhere in the body including visceral organs. NF1-associated glomus tumors also appear to occur mostly in the subungual area and in young women, as in sporadic cases [69]. There are several case reports highlighting the link between multiple lesions and NF1 [70,71].

## 13. Melanoma

NF1 patients have about four-fold higher risk of neoplasms [72] through dysregulation of the RAS/MAPK pathway [73]. Although one of the main drivers in melanoma development is also NF1 mutation, the incidence of melanoma in NF1 patients is slightly elevated [74]. One review concluded that NF1 patients with melanoma were mostly younger-aged women and the tumor thickness was greater [75]. Although NF1 mutations have been associated with desmoplastic melanoma in the general population [76], only one case has been reported as a desmoplastic melanoma but interestingly sharing some morphologic characteristics with MPNSTs [77] (Figure 14).

## 14. Other Neoplasms

NF1 alteration comes along with an increased risk of developing malignant conditions such as MNPST, leukemia, rhabdomyosarcoma and adrenal gland tumors. Pheochromocytoma is a rare benign adrenal medulla tumor that is capable of produce excessive amounts of catecholamine resulting in tachycardia, hypertension, sweating, and flushing. The overall incidence is low—0.1–5.7% of patients with NF develop pheochromocytoma—mostly between the third and fourth decade of life. However, the specific alteration responsible for the association is still unknown. It is usually solitary and benign but long-term follow-up is recommended. Surgical removal is the standard approach for the treatment [78].

Rhabdomyosarcoma (RMS) is the most common soft-tissue sarcoma in children. Sporadic RMS is more common, but RASopathies and disorders with TP53 alterations such as NF1 have also been associated with RMS. Around 1% of patients with NF1 develop RMS, specifically embryonal-type due to RAS activation in this form. RMS in NF1 cases seems to have early onset favoring the pelvic location. Careful clinical examination and attention to any sign pointing out possible RMS are important for early diagnosis. NF1-associated RMS requires the same treatment protocol as the sporadic cases [79].

Recently, several reports have depicted the relationship between gastrointestinal stromal tumors (GISTs) and NF1. NF1-associated-GISTs tend to occur at a younger age in distal locations of the gastrointestinal tract. They generally have lower mitotic indexes and do not show typical KIT/platelet-derived growth factor receptor alpha mutations [80].

## 15. Conclusions

NF1 is a genetic disorder which needs a collaborative and interdisciplinary approach to understand and manage the condition. The future direction of therapies seems to direct cell type-specific growth control pathways with mutation-directed therapeutics in a personalized manner.

## Figures and Tables

**Figure 1 cancers-13-00463-f001:**
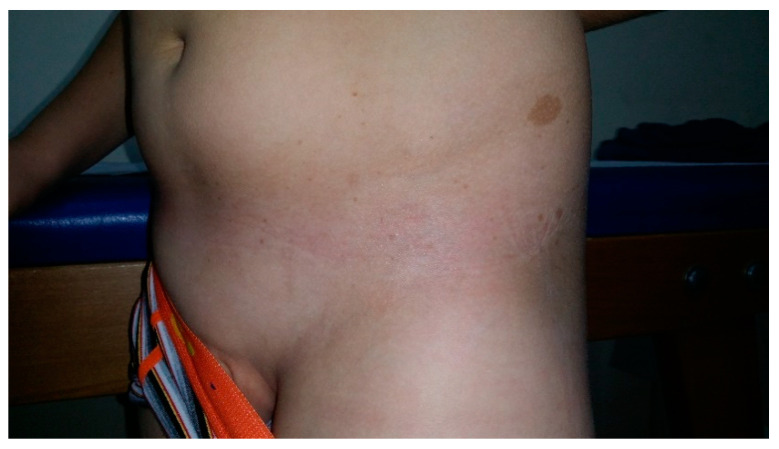
Segmental arrangement of CALMs and freckling in a patient with a mosaic form of NF.

**Figure 2 cancers-13-00463-f002:**
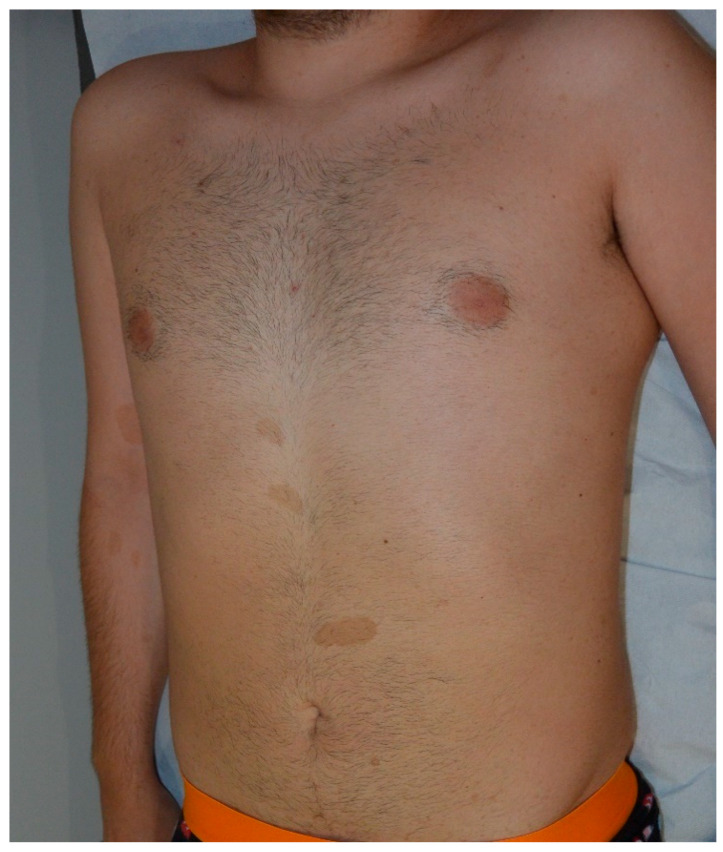
Multiple typical cafe-au-lait macules (CALMs) on the trunk of a young man.

**Figure 3 cancers-13-00463-f003:**
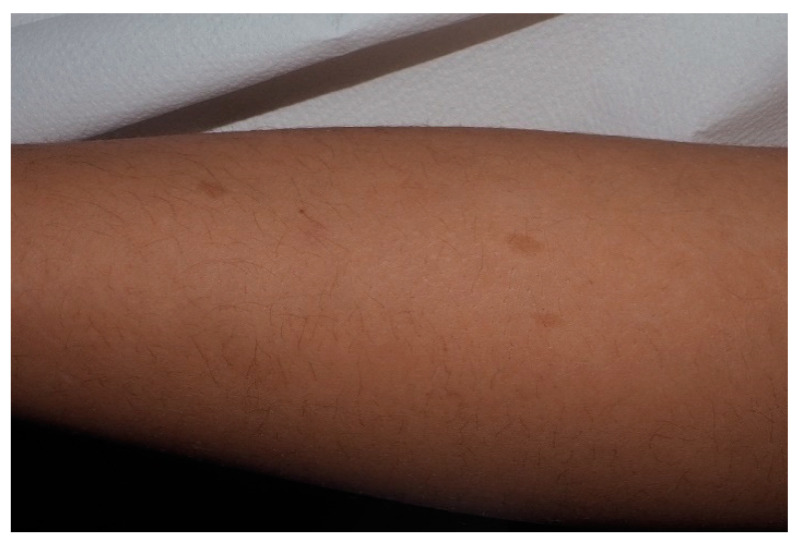
For this child, CALMs were detectable on lower limbs. Note that the size is larger than 5 mm and the typical well-demarcated borders.

**Figure 4 cancers-13-00463-f004:**
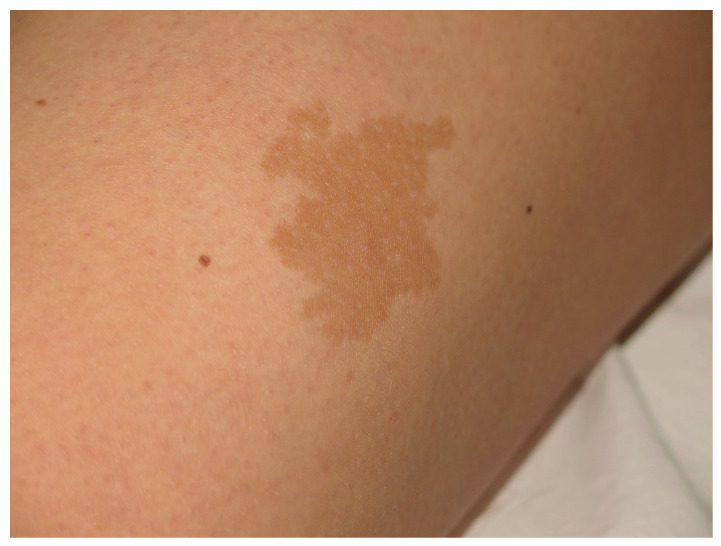
This CALM is not associated with NF1. It shows irregular borders and it is isolated.

**Figure 5 cancers-13-00463-f005:**
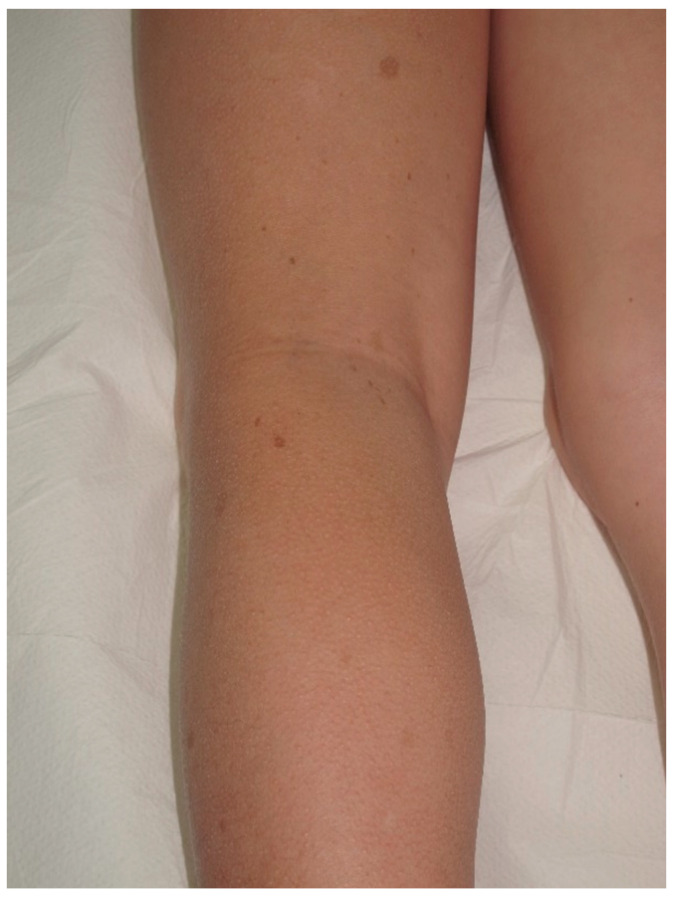
Small CALMs of the left thigh of a young child associated with early flexural freckling.

**Figure 6 cancers-13-00463-f006:**
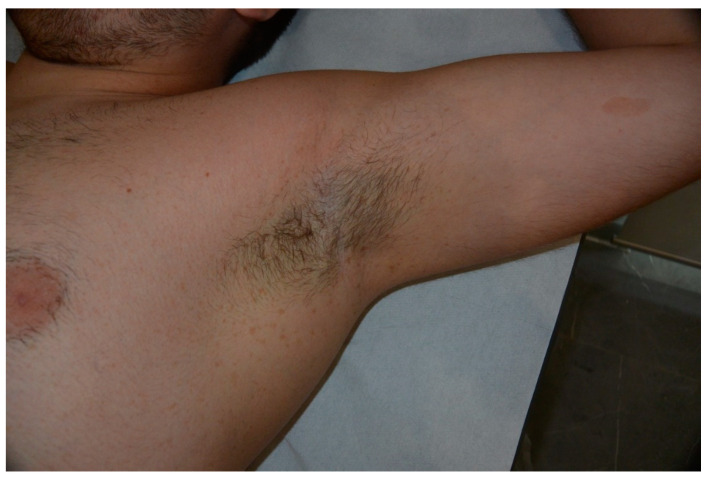
Crowe sign with extensive freckling of axillae in a patient affected by NF1.

**Figure 7 cancers-13-00463-f007:**
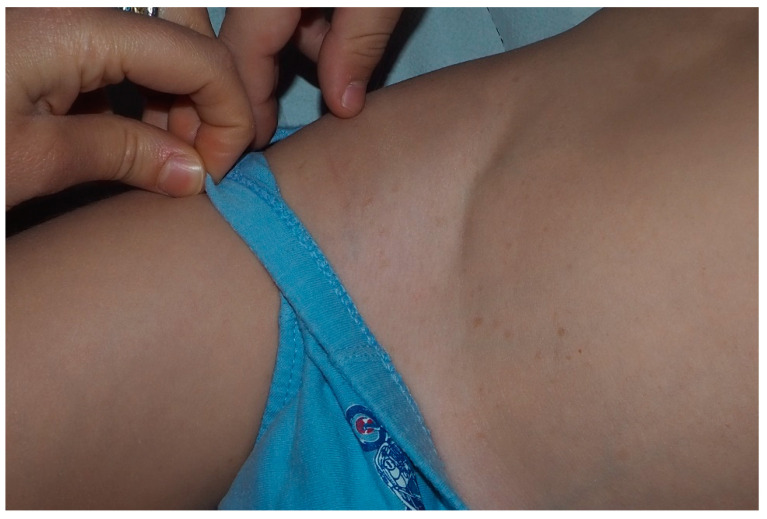
Precocious freckling of inguinal area of a young patient.

**Figure 8 cancers-13-00463-f008:**
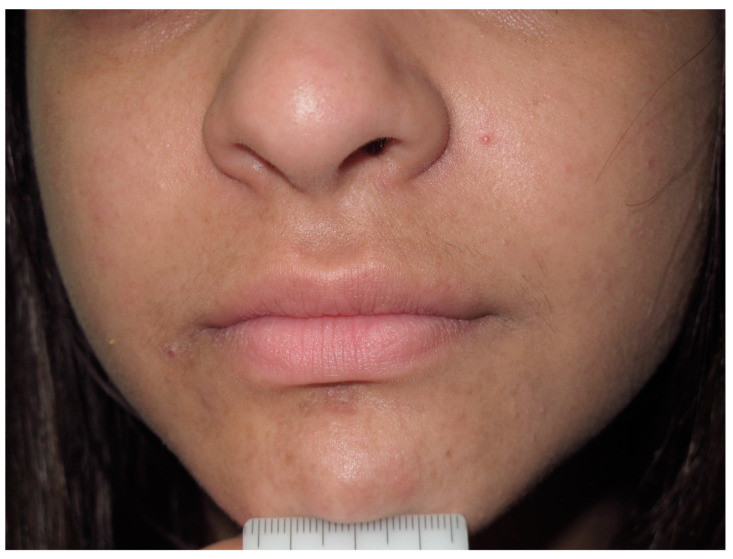
Deep neurofibroma soft of chin.

**Figure 9 cancers-13-00463-f009:**
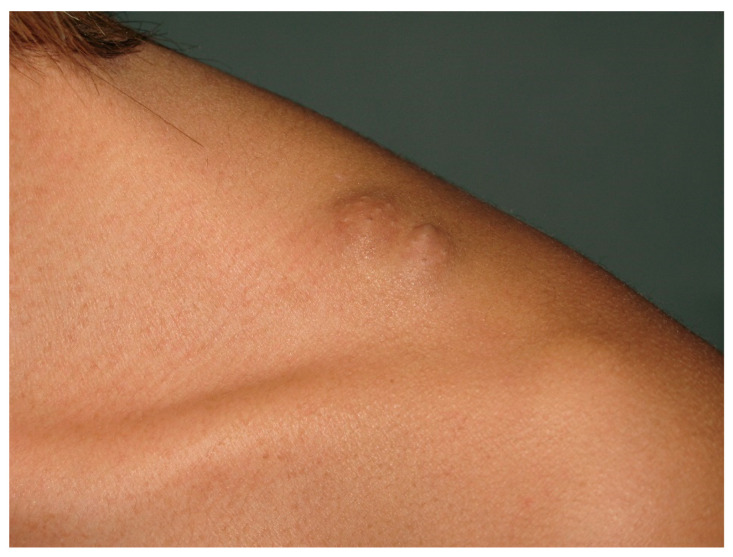
Isolated neurofibroma of supraclavicular region.

**Figure 10 cancers-13-00463-f010:**
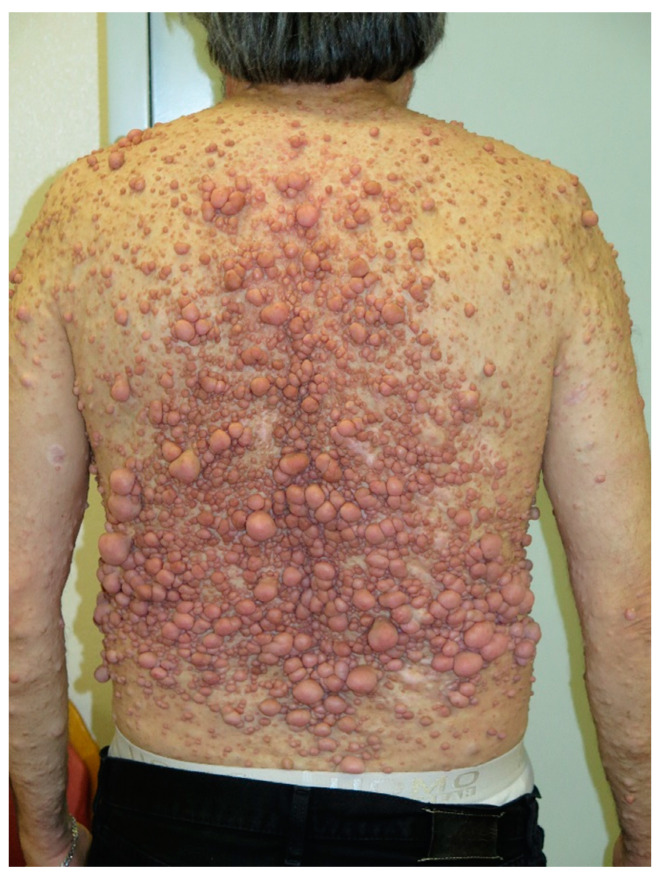
Diffuse neurofibromas on the back.

**Figure 11 cancers-13-00463-f011:**
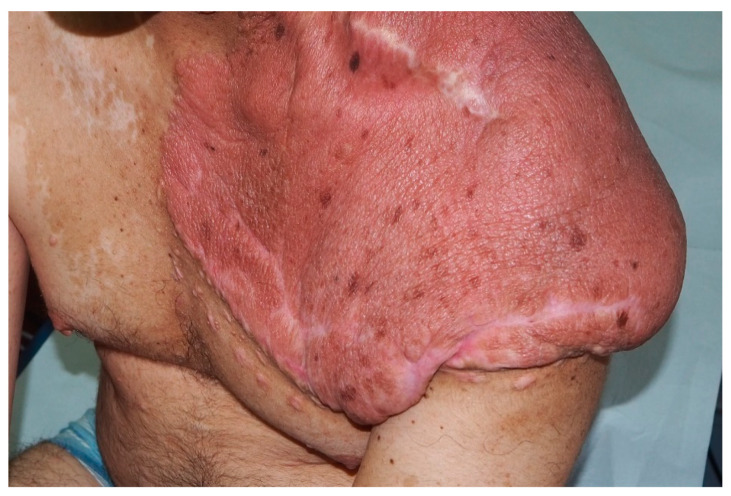
Large plexiform neurofibroma of the left shoulder.

**Figure 12 cancers-13-00463-f012:**
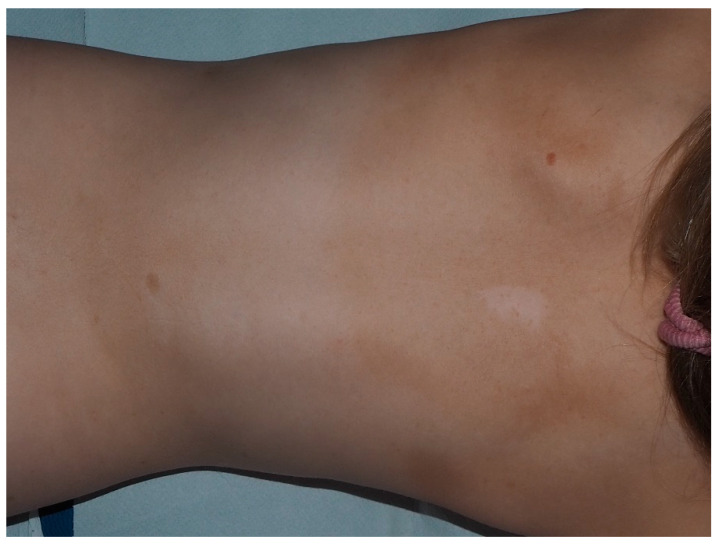
Nevus anemicus of the back in a female with NF1.

**Figure 13 cancers-13-00463-f013:**
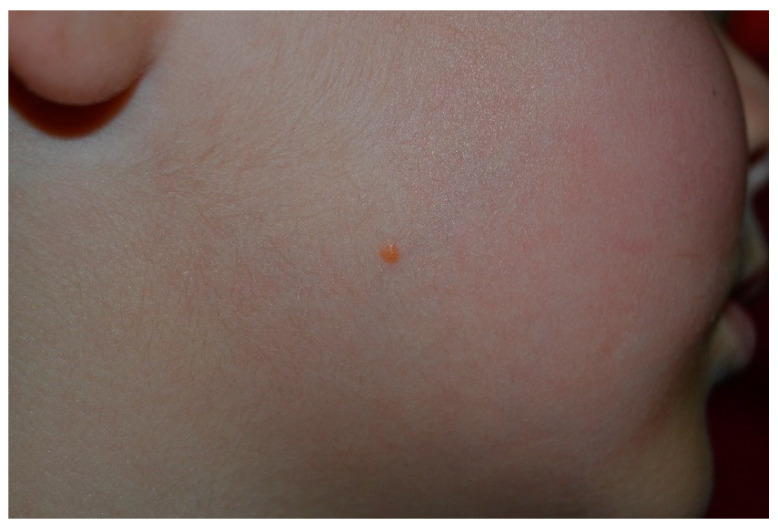
Yellowish papule of the right side of the face, compatible with the diagnosis of juvenile xanthogranuloma.

**Figure 14 cancers-13-00463-f014:**
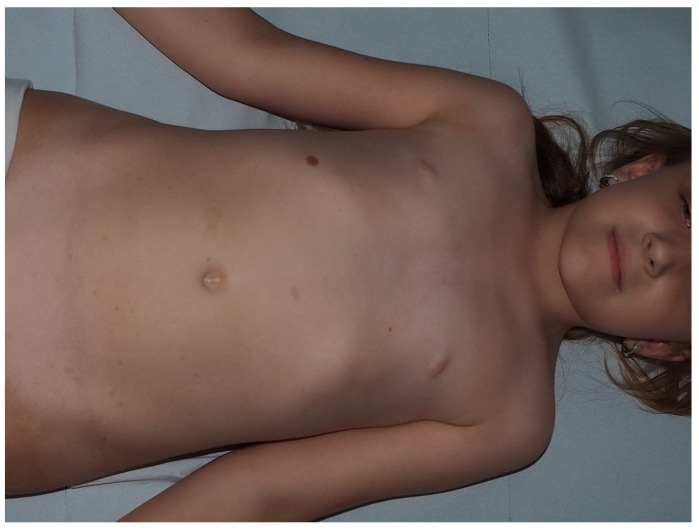
Small congenital nevus associated with multiple CALMs in a girl with NF1.

**Table 1 cancers-13-00463-t001:** Summary of main features of cutaneous findings of Neurofibromatosis type 1 (NF1).

Cafe-au-lait Macules	Light to dark brown-pigmented, oval or round skin patches with regular, well-demarcated margin
Freckles	Small, clustered pigmented lentigines
Neurofibroma	Dome—shaped, pink to brown, pedunculated or sessile cutaneous tumors
Plexiform Neurofibroma	Thickened, slightly raised and pigmeted firm masses located along the nerve; “Bag of worms” appearance; hypertrichosis
Nevus Anemicus	Hypopigmented, pale, well-defined skin patch
Juvenile Xantogranuloma	Yellowish-pink dome-shaped granulomatous papules and nodules
Glomus Tumor	Small reddish-blue papulonodule often located at the subungual area
Melanoma	Unevenly pigmented unusual looking macule

## Data Availability

No new data were created or analyzed in this study. Data sharing is not applicable to this article.

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
