# Peer review of "Cutaneous Findings in Neurofibromatosis Type 1"

_cancers, 2021, doi:10.3390/cancers13030463_

Round 1
Reviewer 1 Report
The authors reviewed NF1. This review is concise, is easy to be read and is instructive.
However, I could find no Figures, although there is 14 figure legends.
Author Response
Thanks for your positive comments.
Figures are available in supplemental material.
Reviewer 2 Report
The authors presented the most common skin lesions associated with neurofibromatosis type 1. Photos of these skin lesions are an important supplement to the work.
However, a few points need to be clarified.
- Judging by the content of the article, the type of work it represents is review than commentary.
- What new information does the study add as compared to previous reviews on this topic. Perhaps a reference to current treatments for this type of skin lesion would add more appeal and novelty to the work.
Author Response
Thanks for your comments.
Information about treatment was added to the manuscript.
Reviewer 3 Report
The author describes dermatologic manifestations associated with NF1 patients. The review is well written.
A few things need to be added
Current review includes a brief information on dermatological manifestations associated with NF1 patients. A brief discussion on NF1 inherited conditions or inheritance pattern, diagnostic testing, current treatment options and future treatment possibilities with proper citations may provide a detailed information for the readers.
Author should briefly discuss about benign and cancerous tumors such as Lisch nodules, glioma, MPNST or sarcoma, spinal cord tumor or adrenal gland tumor that occurs in people with NF1 and the associated features or health ailments in these patients.
Cafe-au-lait macules (CALMs), after first abbreviation, CALM should be used in whole manuscript.
Author Response
Thanks for useful comments. 1- We added a title about the genetic background. Under this heading, we have touched upon the inheritance pattern, diagnostic criteria, genetic diagnostic test and genotype-phenotype correlations. 2- We detailed the title of Neurofibroma. We talked about available treatments and clinical trials. I specifically mentioned the Spinal NF1. 3- We made malignant peripheral nerve sheath tumors a separate title. We talked about the genetic background and treatment in more detail. 4- Lisch nodules and optic glioma were treated as separate titles. 5- We mentioned NF1-associated neoplasms; such as sarcoma, adrenal gland tumors, gastrointestinal tumors. 6-Finally, since the word acount exceed 4000, very short conclusion part has been added as well. 7- CALM has been used in the manuscript as acronymRound 2
Reviewer 2 Report
The manuscript has been revised according to all the reviewer suggestions and comments.